# Phenotypic and Genotypic Identification of Dermatophytes from Mexico and Central American Countries

**DOI:** 10.3390/jof9040462

**Published:** 2023-04-11

**Authors:** Angélica Pérez-Rodríguez, Esperanza Duarte-Escalante, María Guadalupe Frías-De-León, Gustavo Acosta Altamirano, Beatriz Meraz-Ríos, Erick Martínez-Herrera, Roberto Arenas, María del Rocío Reyes-Montes

**Affiliations:** 1Departamento de Microbiología y Parasitología, Facultad de Medicina, Universidad Nacional Autónoma de México (UNAM), Ciudad Universitaria No. 3000, Mexico City 04510, Mexicobmerazr@hotmail.com (B.M.-R.); 2Hospital Regional de Alta Especialidad de Ixtapaluca, Carretera Federal México-Puebla Km. 34.5, Pueblo de Zoquiapan, Ixtapaluca 56530, Mexicomq9903@live.com.mx (G.A.A.); 3Sección de Estudios de Posgrado e Investigación, Escuela Superior de Medicina, Instituto Politécnico Nacional, Plan de San Luis y Díaz Mirón, Mexico City 11340, Mexico; 4Departamento de Dermatología, Sección de Micología, Hospital General Dr. Manuel Gea González, Mexico City 10480, Mexico

**Keywords:** dermatophytes, fungi, molecular marker, multilocus sequence typing, phylogeny

## Abstract

Dermatophytes are fungi included in the genera *Trichophyton*, *Microsporum*, *Epidermophyton*, *Nannizzia*, *Paraphyton*, *Lophophyton*, and *Arthroderma*. Molecular techniques have contributed to faster and more precise identification, allowing significant advances in phylogenetic studies. This work aimed to identify clinical isolates of dermatophytes through phenotypic (macro- and micromorphology and conidia size) and genotypic methods (sequences of ITS regions, genes of β tubulin (*BT2*), and elongation factor α (*Tef-1α*)) and determine the phylogenetic relationships between isolates. Ninety-four dermatophyte isolates from Costa Rica, Guatemala, Honduras, Mexico, and the Dominican Republic were studied. The isolates presented macro- and micromorphology and conidia size described for the genera *Trichophyton*, *Microsporum*, and *Epidermophyton*. Genotypic analysis classified the isolates into the genera *Trichophyton* (63.8%), *Nannizzia* (25.5%), *Arthroderma* (9.6%), and *Epidermophyton* (1.1%). The most frequent species were *T. rubrum* (26 isolates, 27.6%), *T. interdigitale* (26 isolates, 27.6%), and *N. incurvata* (11 isolates, 11.7%), *N. gypsea* and *A. otae* (nine isolates, 9.6%), among others. The genotypic methods clarified the taxonomic status of closely related species. For instance, the ITS and BT2 markers of *T. rubrum*/*T. violaceum* did not differ but the *Tef-1α* gene did. On the other hand, the three markers differed in *T. equinum*/*T. tonsurans*. Therefore, the ITS, *BT2*, and *Tef-1α* genes are useful for typing in phylogenetic analyses of dermatophytes, with *Tef-1α* being the most informative locus. It should be noted that isolate MM-474 was identified as *T. tonsurans* when using ITS and *Tef-1α*, but when using *BT2*, it was identified as *T. rubrum*. On the other hand, no significant difference was found when comparing the methods for constructing phylogenies, as the topologies were similar.

## 1. Introduction

Dermatophytosis is a superficial skin infection caused by a closely related group of filamentous fungi called dermatophytes with the ability to digest and grow in keratinized structures such as skin, nails, hair, claws, horns, and feathers [1]. These fungi generate mild lesions in immunocompetent hosts, whereas in immunocompromised hosts, they cause severe and disseminated infections [2]. Dermatophytes present a worldwide distribution, being relevant in veterinary and public health [3]. Transmission of dermatophytes occurs through direct contact with infected people, animals, or contaminated objects (fomites) [4].

In their anamorphic state, these fungi mainly belong to the genera *Trichophyton*, *Microsporum*, and *Epidermophyton*. However, the new taxonomy proposed by de Hoog et al. [5] integrates four new genera: *Arthroderma*, *Lophophyton*, *Nannizzia*, and *Paraphyton*. The teleomorph state of these fungi integrates the genera *Arthroderma* and *Nannizzia* [6]. Based on their habitat, dermatophytes are classified as anthropophilic, zoophilic, or geophilic.

Until recently, phenotypic methods, including morphology, physiology, and biochemistry, were the basis of dermatophyte taxonomy and identification. However, in many cases, more is needed to identify less-common taxa or new species [7]. Identifying dermatophytes at the species level is necessary, as they can show different susceptibility patterns to antifungal agents. The latter has been demonstrated in antifungal susceptibility tests with isolates of *T. rubrum*, *T. tonsurans*, and *T. equinum* in vitro [8,9,10]. It has also been observed that traditional diagnostic tests can be unstable and imprecise, which is why molecular methods have been implemented. Molecular methods allow identification at the genus and species level and complement or replace conventional methods [11,12,13,14].

DNA sequencing and molecular systematics shaped a new concept of species in dermatophytes, the phylogenetic species, which reduces the number of recognized taxa [12]. Furthermore, they have led to the discovery of new cryptic species [7,14,15]. Consequently, significant advances have been made in modern systematics during the last decade using molecular markers such as internal transcribed spacer regions 1 and 2 (ITS1 and ITS2) of rDNA [12]. These markers are helpful when examining taxonomically related organisms [16,17,18]. The ITS markers and the oligonucleotide system designed by Makimura et al. [19] and White et al. [20] have been widely used in phylogenetic studies [19,21,22,23]. Likewise, the sequence of a gene segment that codes for β-tubulin has been advantageous for species delimitation in other groups of fungi, such as *Aspergillus*, *Penicillium*, *Scedosporium*, and *Phaeoacremonium* [24,25,26,27]. Rezaei-Matehkolaei et al. [2] used the BT2 marker and the ITS region to assess intra- and interspecific dermatophyte variations and found better resolution with the *BT2* marker.

The sequence of the elongation factor 1-α (*Tef-1α*) gene, which encodes a part of the protein translation machinery, was first used as a marker for *Fusarium* [28,29]. This marker has also been evaluated in the identification of dermatophytes. Mirhendi et al. [30] showed high consistency between the phylogeny obtained with the ITS and Tef-1α markers. The latter showed greater discriminatory power for related species, such as *A. vanbreuseghemii*, *T. rubrum*, *A. benhamiae*, and *A. otae*. However, the authors emphasize that an individual marker cannot specify the limits between dermatophyte species. In contrast, multilocus markers allow for a more precise evaluation of the relationships between isolates.

It is known that some species of dermatophytes have geographically delimited areas to a greater or lesser extent [31]. It has been shown that both the appearance of species and reductions in the number of species are associated with habitat changes and increased mobility of people from continent to continent. Philpot [31] showed that species related to clinical forms vary according to geographic origin. In recent years, changes have been observed in the epidemiology of mycoses associated with changes in clinical patterns, for example, environmental changes, dispersion of etiological agents, a surge of HIV cases, use of immunosuppressive therapies, and increased resistance to antifungals due to their indiscriminate use, among others [32,33].

Therefore, insight into the geographic distribution of these pathogens is crucial when making a diagnosis, together with phenotypic and genotypic identification of the species. Unfortunately, published works on the epidemiological data of dermatophytosis in Mexico and Central America are scarce, and the diagnosis is restricted to phenotypic methods. Therefore, it is convenient to combine conventional and molecular methods to identify species of dermatophytes and determine their relationship with their geographical origin [34,35,36]. Thus, this work aimed to identify the species of clinical isolates from Costa Rica, Guatemala, Honduras, Mexico, and the Dominican Republic using phenotypic and genotypic methods and to construct their molecular phylogeny.

## 2. Materials and Methods

### 2.1. Clinical Samples

Ninety-four clinical skin, nails, and hair samples were used to isolate dermatophytes. Among them, 55 isolates from Guatemala were provided by the Institute of Dermatology and Skin Surgery “Prof. Dr. Fernando A. Cordero C.” in Guatemala City; 32 isolates from Mexico and three from Honduras were provided by the Hospital General Dr. Manuel Gea González in Mexico City; three isolates from the Dominican Republic were obtained from the Dermatology and Skin Surgery Institute “Dr. Huberto Bogaert Díaz” in Santo Domingo; and one isolate from Costa Rica was obtained from the Golfito Hospital Manuel Mora Valverde in San José (Table 1).

### 2.2. Phenotypic Characterization

Isolating dermatophytes. The dermatophyte samples from Costa Rica, Guatemala, Honduras, Mexico, and the Dominican Republic were sown on Sabouraud agar with cycloheximide and chloramphenicol (soja peptone 10.0 g/L, dextrose 10 g/L, agar 15.5 g/L, cycloheximide 0.4 g/L, and chloramphenicol 0.105 g/L) (Bioxon, Mexico City, Mexico) and incubated at 28 °C for two weeks or until the growth of filamentous fungi was observed. Initially, the cultures were observed using the Rush−Munro technique. Next, the preparation was observed under an optical microscope with 10× and 40× objectives to identify the characteristic structures of dermatophytes.

Obtaining monosporic cultures. Monosporic cultures were obtained based on the methodology described by Valencia-Ledezma et al. [37].

Macromorphology. The isolates were seeded in Petri dishes with potato dextrose agar (PDA) medium (potato 4 g/L, dextrose 20 g/L, and agar 15 g/L) (Bioxon) and Sabouraud agar with cycloheximide and chloramphenicol (Bioxon) and incubated at 28 °C for 4−8 days. Subsequently, the morphological characteristics of the colonies, including color, surface appearance, and pigmentation, were observed. The isolates were photographed with a digital camera (Cyber-Shot, 8.1 megapixels, Sony, Mexico City, Mexico) for documentation purposes.

Micromorphology. The dermatophytes’ micromorphological features were analyzed using Ridell’s microculture technique [38]. The microscopic characteristics of the isolates were recorded with a digital camera (Sony).

Conidia size. Depending on the fungus species, 30 microconidia, ten macroconidia, or both were measured from each microculture obtained, according to the methodology described by Frías de León et al. [39].

### 2.3. Genotypic Characterization

Genomic DNA extraction. Each monosporic dermatophyte culture seeded in Sabouraud with cycloheximide and chloramphenicol (Bioxon) was inoculated in 50 mL yeast extract peptone dextrose (YEPD) liquid medium (66.66% dextrose, 16.67% yeast extract, 16.67% casein peptone) and incubated at 37 °C under stirring for two days or until mycelial growth was observed. The mycelial biomass of each isolate was harvested by filtration and frozen at −20 °C until use. The fungal cell wall was initially broken to obtain DNA according to the method proposed by Williams et al. [40]. Subsequently, a DNeasy^®^ Plant mini kit (Qiagen, Austin, TX, USA) was used, following the manufacturer’s recommendations. The total extracted DNA was quantified by a qualitative method through 1% agarose gel electrophoresis and compared with different concentrations (10, 30, and 50 ng/µL) of phage λ (Gibco BRL^®^, San Francisco, CA, USA) and stained with GelRed™ 10,000× Biotium (Fremont, CA, USA). It was also measured quantitatively by UV spectrophotometry using a NanoDrop 2000 spectrophotometer (Thermo Fisher Scientific, Waltham, MA, USA).

Genotypic typing. Three molecular markers were used: (1) a partial sequence of the β-tubulin gene (*BT2*) amplified with oligonucleotides T1-F and Bt2b, described by Salehi et al. [41] and Glass and Donaldson [42]; (2) a sequence of the ITS region amplified with the ITS1-F and ITS4 oligonucleotides, reported by Taghipour et al. [43]; and (3) a partial sequence of the elongation factor 1-α (*Tef-1α*) gene amplified with oligonucleotides EF-DermF and EF-DermR, described by Salehi1 et al. [41].

The PCR products were analyzed by electrophoresis on a 1.5% agarose gel stained with GelRed™ 10,000× Biotium (Thermo Fisher Scientific). Electrophoresis was performed at 100 V for 60 min in 0.5X TBE buffer (62.66% Tris base, 31.90% boric acid, 5.44% EDTA). The molecular size standard used was 100 bp DNA Ladder (Invitrogen, New York, NY, USA). Images of the gels were captured on a MultiDoc-It™ Imaging System (Upland, CA, USA).

Sequencing of amplified fragments. The PCR products obtained from all dermatophyte isolates were sent for two-way sequencing (Macrogen Inc., Seoul, South Korea). Electropherograms of the obtained sequences were edited with the BioEdit program v. 7.2.5 [44].

Sequence analysis. The sequences obtained with each marker were compared with all nucleotide sequences belonging to fungi deposited in GenBank (URL4) through the program BLASTn (www.blast.ncbi.nlm.nih.gov/blast.cgi, accessed on 20 October 2022) [45]. In addition, the sequence alignments of the ITS region and the *BT2* and *Tef-1α* genes were analyzed, considering the percentages of similarity, identity, and expectation to corroborate the homology between the obtained sequences.

Phylogenetic analysis. Reference sequences were chosen from each dermatophyte species, corresponding to the ITS region and the *BT2* and *Tef-1α* genes, to perform the phylogenetic analysis using the maximum likelihood and Bayesian inference methods. The MEGAX program (www.megasoftware.net/, accessed on 2 November 2022) was used to obtain a tree through maximum likelihood [46], and MrBayes 3.2.2 was used to obtain a tree through Bayesian inference (//mrbayes.sourceforge.net/download.php, accessed on 5 November 2022) [47]. In the former, the support of the branches was calculated using bootstrapping with 1000 repetitions [48], whereas the latter was validated through the posterior probability values [49]. The phylogenetic trees were edited using FigTree v. 7.2.5 [50]. *Myceliophthora lutea* (access nos. KM655312.1 (ITS), KX977026.1 (*BT2*), and HQ871722.1 (*Tef-1α*)) were used as an outgroup in both phylogenetic trees.

The sequences obtained with the *BT2*, *Tef-1α*, and ITS markers were deposited in GenBank (access numbers: OQ319835-OQ319928 (ITS), OQ344337-OQ344430 (*BT2*), and OQ414474-OQ414567 (*Tef-1α*)).

## 3. Results

### 3.1. Phenotypic Characterization

Isolation and identification of dermatophytes. Ninety-four dermatophyte isolates were obtained (Table 1). The isolates, identified through the Rush−Munro technique (adhesive tape), showed particular structures, which allowed them to be grouped into the genera *Trichophyton*, *Microsporum*, and *Epidermophyton*.

According to the macro- and micromorphological characteristics of each isolate, the following species were identified: *T. rubrum* (27), *T. mentagrophytes* (27), *M. gypseum* (21), *M. canis* (9), *T. tonsurans* (6), *N. nana* (3), and *E. floccosum* (1) (Appendix A).

Conidia size. Table 2 shows the average conidia size of each dermatophyte species identified by phenotypic methods. It should be noted that the average size was calculated by measuring 30 microconidia, ten macroconidia, or both, depending on the species.

### 3.2. Genotypic Characterization

The sequences obtained with each marker (ITS, *BT2*, and *Tef-1α*) were edited and used to build phylogenetic trees through the maximum likelihood and Bayesian inference methods.

Maximum likelihood. The phylogenetic tree inferred from the ITS region sequences showed eight clusters (Appendix A). Similarly, the phylogenetic tree constructed from the *BT2* gene sequences also formed eight groups (Appendix A). The phylogenetic tree generated from the *Tef-1α* gene sequences showed nine groups (Appendix A). Generally, the topologies with the three markers using the maximum likelihood method showed similarities.

A tree with concatenated sequences was built to obtain more information from the three markers. Nine groups were distinguished in this tree (Figure 1). Table 3 shows the results of the phylogeny obtained through the maximum likelihood method and the identity of the isolates studied in each group associated with the related species.

Bayesian inference. The phylogenetic tree inferred from sequences of the ITS region showed nine groups (Appendix A). Similarly, the phylogenetic tree constructed from sequences obtained with the *BT2* marker formed nine groups (Appendix A). Finally, the phylogenetic tree obtained from *Tef-1α* gene sequences showed eight groups (Appendix A). Overall, the topologies with the three markers using Bayesian inference showed similarities.

Likewise, a tree was built with concatenated sequences to obtain more information from the three nuclear genes. Nine groups were distinguished in the tree (Figure 2). Table 4 shows the phylogeny results obtained by Bayesian inference and the identity of the isolates studied in each group associated with the related species.

The results regarding the frequency of species identified in this study using the concatenated sequences of the three markers were similar between the maximum likelihood and Bayesian inference methods, as shown in Table 5 and Table 6. The most frequent species in Mexico was *T. violaceum*/*T. rubrum*, whereas in Guatemala, the most frequent species were *T. mentagrophytes*/*T. interdigitale*, *N. incurvata*, and *M. gypseum*/*N. gypsea*.

## 4. Discussion

According to the World Health Organization (WHO), dermatophytosis affects around 25% of the world’s population, and its incidence is higher in tropical areas due to the high temperature and humidity [50]. The rise in species that cause dermatophytosis can vary depending on geographical location, migration patterns on different continents, changes in human lifestyles, and the medical approach to dermatophytosis in different health systems; all these factors influence the predisposition to dermatophyte infection [51].

In Mexico and Central American countries, dermatophytes are generally identified using phenotypic methods. However, genotypic methods are required for accurate species identification because the taxonomy of dermatophytes has changed significantly in recent years. Therefore, in this work we analyzed 94 clinical samples of dermatophytes from Costa Rica, Guatemala, Honduras, Mexico, and the Dominican Republic, and identified them by using phenotypic and genotypic methods.

It is known that dermatophytes present significant variability in their macroscopic morphology, such as pigment production, appearance, and consistency of the colony. In addition, their micromorphological diversity includes typical conidial arrangement and different hyphal modalities. All of these are essential characteristics for identifying the species. In the present work, we used APD and Sabouraud culture media with cycloheximide and chloramphenicol to reveal dermatophytes’ macro- and micromorphological variability. These culture media also favored visualization of the pigments produced by some of the studied isolates.

The micromorphology of isolates in the PDA medium showed various structures in the vegetative and reproductive mycelia; for example, pyriform microconidia predominated in *T. rubrum*, whereas globose microconidia were abundant in species of the *T. mentagrophytes* complex. Some isolates showed pyriform and globose microconidia in the same proportions, so this characteristic could have been more valuable in differentiating between species of the same complex. Similarly, isolates identified as *T. tonsurans* showed numerous conidia when they were sown in the PDA medium, contrary to what was observed when they were sown in the Sabouraud medium with cycloheximide and chloramphenicol.

*N. gypsea* and *N. nana* isolates shared morphological characteristics, such as abundant colony growth and a powdery appearance, with most showing buff or brown colonies. *M. canis* isolates developed cottony white colonies with limited growth. Isolates of *N. gypsea* and *M. canis* showed spindle-shaped macroconidia and those of *N. nana* were pear-shaped. Thus, the main difference between the three species was the number of locules on their macroconidia: *M. canis* had ten locules, *N. gypsea* had 3−6 locules, and *N. nana* had two locules. In addition, the *E. floccosum* isolate grown in the PDA medium showed abundant rod-shaped macroconidia; however, in the Sabouraud medium with cycloheximide and chloramphenicol, abundant chlamydoconidia were observed. It is worth noting that one of the main characteristics of this species is the absence of microconidia.

Isolates MM-406, MM-459, MM-475, and MM-491 showed pleomorphism; therefore, only their colonial morphology was described, highlighting the importance of genotypic identification. In addition, it has been proven that the methods used for morphological identification of dermatophytes are insufficient and prone to error [52], as is the case for *T. interdigitale* and *T. rubrum*, which are very phenotypically similar [53,54,55].

Furthermore, by using several markers, molecular analysis has allowed for a better understanding of the taxonomic and phylogenetic relationships during the last few years. For instance, the nuclear rDNA region has been very advantageous in phylogenetic studies, because it is multicopy and contains highly conserved genes such as 18S, 5.8S, and 28S, as well as the variable domains of ITS1 and ITS2 and the non-transcribed IGS region. The 18S and 28S regions often harbor introns inserted at highly conserved positions, providing more information on this region. The properties of these markers have been extensively exploited in studies of *Aspergillus* species [55,56,57,58] and other fungi [59,60,61]. The ITS region was the first used to construct phylogenies for dermatophyte identification, allowing an advance in their taxonomy. However, the species have been confirmed and distinguished using other genes. The *BT2* marker has also been used to construct a dermatophyte phylogeny, showing a tree with a topology similar to that obtained with the ITS marker, with minor inconsistencies [2]. Another marker is *Tef-1α*, which has been compared with the ITS marker. The results showed consistency between the topologies of both trees; however, the specificity and discriminatory power were higher with Tef-1α than with ITS, which is particularly useful in some closely related species groups [30]. Thus, considering the characteristics and usefulness of the *BT2*, *Tef-1α*, and ITS markers, we included them in the genotypic characterization of dermatophytes in the present study.

This study clearly shows the advantage of using genotypic over phenotypic methods to accurately identify dermatophytes, as only seven species were identified phenotypically (*T. rubrum*, *T. mentagrophytes*, *M. gypseum*, *M. canis*, *T. tonsurans*, *N. nana*, and *E. floccosum*), whereas 11 species were identified genotypically (*T. rubrum*, *T. mentagrophytes*, *N. gypsea*, *T. tonsurans*, *N. nana* and *E. floccosum*, *T. interdigitale*, *N. fulva*, *N. incurvata*, *A. otae*, and *T. equinum*).

It is vital to consider that constructing a fungal phylogeny based on morphological criteria or with individual genes, such as rRNA gene sequences, does not always determine with certainty the taxonomic level of the examined organisms [62]. In addition, studies based on a single marker only sometimes faithfully represent the history of the entire genome of an organism, and comparisons can give erroneous conclusions about the organism’s relationship with members of the same species or even the same genus [63]. Therefore, this study used the sequences obtained with three markers (*BT2*, *Tef-1α*, and ITS) to construct an individual phylogeny for each marker. A phylogeny was also obtained with the concatenated sequences of the three markers.

Using the maximum likelihood method, this study used the ITS, *BT2*, and *Tef-1α* genes to construct phylogenies. With these genes, different levels of resolution of the groups were obtained. The numbers of clusters supported with bootstrap values of >70% for *Tef-1α* > ITS > *BT2* were observed to be nine, eight, and eight, respectively (Appendix A). Moreover, the phylogeny obtained using maximum likelihood with the three concatenated sequences showed nine groups with bootstrap values of >70%.

The ITS, *BT2*, and *Tef-1α* genes were also used to construct phylogenies using the Bayesian inference method. Different levels of resolution of the groups were obtained using these genes. Regarding the number of groups supported with posterior probability values of 0.7−1, it was observed that ITS > *BT2* > *Tef-1α* yielded nine, nine, and eight groups, respectively (Appendix A). The phylogeny obtained using Bayesian inference with the three concatenated sequences showed nine groups with posterior probability values of 0.7−1.

The comparison between Bayesian inference and maximum likelihood using the concatenated sequences of the three markers showed that Bayesian inference was more valuable for discriminating the species of the genus *Trichophyton* and the main species of the genus *Microsporum*. However, the analyses showed that *T. tonsurans* and *T. equinum* were in the same group, forming two subgroups. With the markers used, it was impossible to separate these species because they have such a close relationship due to the fact that they are a complex species, so it is proposed that *T. equinum* is derived from *T. tonsurans* [41].

It should be noted that, in all trees, *E. floccosum* was kept separate from the other genera. A closer relationship with the genus *Microsporum* is shown, especially for the species *N. nana*. These results agree with Rezaei-Matehkolaie et al. [2], who noted that the genus *Epidermophyton* formed a group close to *Microsporum* when the ITS and *BT2* markers were used. However, some works suggest that *E. floccosum* is closely related to the anthropophilic species of *Trichophyton* [64]. It was also observed that with the three markers and two phylogenetic methods, the species *T. rubrum* and *T. violaceum* showed molecular similarity. These results agree with those reported by Zhan et al. [65], who showed that these two species were phenotypically different but highly similar phylogenetically, because their multilocus phylogeny (with five markers) and a comparison of their genomes show close affinity. Therefore, the possibility that they represent a single species has been discussed. However, they present different phenotypes due to different locations in the human body (*T. violaceum* has a predilection for the scalp, whereas *T. rubrum* is found in the skin and nails). Alternatively, we can hypothesize that *T. violaceum* is simply a phenotypically different strain from *T. rubrum* that has arisen due to differences in physiological stress occurring in different habitats, because it is crucial to consider that virulence and adaptation are also essential for definition of the species when determining the correct affiliation of the species and the limits between them. In addition, the ITS and *Tef-1α* markers identified the MX55 isolate as *T. tonsurans*, whereas the *BT2* marker identified it as *T. rubrum*. Nevertheless, when performing the concatenated analysis, defining a group was impossible. Therefore, it is proposed that this isolate pertains to a new genotype.

Furthermore, relevant epidemiological data were obtained in this work. Using the maximum likelihood method, the most frequent species in Mexico were found to be *T. violaceum*/*T. rubrum*, *T. equinum*, *T. tonsurans*, *T mentagrophytes*/*T. interdigitale*, and *E. flocosum*. In Guatemala they were *T. mentagrophytes*/*T. interdigitale*, *N. incurvata*, *M. canis*, *M. gypseum*/*N. gypsea*, *N. nana*, and *T. violaceum*/*T. rubrum*, whereas in Honduras one isolate corresponded to *T mentagrophytes*/*T. interdigitale*, another to *T. violaceum*/*T. rubrum*, and a third to *N. fulva*. Regarding the isolates from the Dominican Republic, one corresponded to *T. mentagrophytes*/*T. interdigitale*, another to *T. tonsurans*, and a third to *T. violaceum*/*T. rubrum*, and the only isolate from Costa Rica corresponded to *T. violaceum*/*T. rubrum*. Using the Bayesian inference method, the most frequent species in Mexico were found to be *T. violaceum*/*T. rubrum*, *T. equinum*, *T. tonsurans*, *T mentagrophytes*/*T. interdigitale*, and *E. floccosum*. In Guatemala they were *T. mentagrophytes*/*T. iterdigitale*, *N. incurvata*, *N. gypsea*, *M. canis*, and *N. nana*. In Honduras, one isolate corresponded to *T. violaceum*/*T. rubrum*, another to *T. mentagrophytes*/*T. interdigitale*, and one to *N. fulva*. In the Dominican Republic, one corresponded to *T. violaceum*/*T. rubrum*, another to *T. tonsurans*, and one to *T. mentagrophytes*/*T. interdigitale*, and the only isolate from Costa Rica corresponded to *T. violaceum*/*T. rubrum*. These findings agree with data reported by Borman et al. [66], who reviewed global trends of dermatophytes in the last three decades and showed that all reported *T. mentagrophytes* were considered as *T. interdigitale*. For this reason, they noted that *T. rubrum* is one of the main etiological agents in most developed countries, followed by *T. interdigitale* in central and southern Europe and *T. tonsurans* on the American continent.

In addition, it has been shown that the prevalence of *N. gypsea* worldwide is low. For example, in Siena, Italy, this species represents 6.8% of all infections caused by dermatophytes [67]; however, in Brazil, the frequency ranges from 0.8 to 2.5% [68]. However, the results obtained in the present study differ significantly, with *N. gypsea* representing 9.6% of the total isolates. On the other hand, Ebrahimi et al. [69] analyzed 79 clinical samples from Mashhad, Iran, and found that *M. canis* represented 10.1% of all causal agents. It has also been reported that on the American continent, its frequency is 7.0% among the most common mycoses [70], which agrees with our results.

Spiewak and Szostak [71] evaluated the frequency of dermatophyte infections in 190 farmers from 1997 to 2000 and identified one case caused by *N. nana*, representing 0.53% of the total identified species. Saghrouni et al. [72] conducted a 26-year retrospective study (1983–2008) on the epidemiological aspects of *Tinea capitis* in the Sousse region (central Tunisia). The authors analyzed 5562 cases, among which *N. nana* represented 0.01% of the etiological agents. However, the percentage of this species obtained in this study was significantly higher, 3.2%. In addition, we only identified one case of *E. floccosum* (1.1%), which differs from some publications that reported a higher frequency, ranging from 31.4 to 32.8% in Tehran and Qazvin, Iran, respectively [73,74].

De Mata and Mata [75] reported that one of the most frequently isolated species among dermatophytes in Costa Rica was *T. rubrum*, followed by *N. gypsea*. Mata and Mayorga [76] also determined that *N. gypsea* was among the predominant causal agents in this country. The present work identified the only isolate from Costa Rica as *T. violaceum*/*T. rubrum*, which agrees with the above results. Nevertheless, it is necessary to analyze a larger number of isolates to corroborate this finding.

Regarding isolates from Guatemala, interestingly, the data do not coincide with what was reported by Martínez et al. [77], who noted that the most common dermatophyte was *T. rubrum*. Likewise, Frías de León et al. [78] investigated the epidemiological status of *Tinea capitis* in a subtropical region of Mesoamerica (Guatemala) over the last 12 years. They found that the most frequently isolated dermatophytes were *M. canis* (82%), *N. gypsea* (6%), and *T. rubrum* (5%).

The three isolates from Honduras were identified as *N. fulva*, *T. violaceum*/*T. rubrum*, and *T. mentagrophytes*/*T. interdigitale*, which partly agrees with the data obtained by Mejta de Calona et al. [79], who reported that *T. rubrum*, *M. canis*, and *T. mentagrophytes* were the most frequent agents causing dermatomycosis. In addition, García and Meléndez [80] analyzed 110 cases of *Tinea capitis* in children under 18 years of age and reported that the most frequent etiological agents were *M. canis* and *T. tonsurans*.

On the other hand, the results for the isolates from Mexico partly agree with those of Welsh et al. [81], who evaluated the frequency of clinical variants and the distribution of etiological agents in the state of Monterrey, Mexico, and showed that the predominant agent was *T. rubrum* (45%). However, in that study the second most frequent agent was *T. mentagrophytes*, whereas in our analysis, *T. tonsurans* ranked second. López Martínez et al. [82] examined 15,101 biological samples from Mexico City and found that the most common species was *T. rubrum* (71.2%), followed by *T. tonsurans* (6.9%), *T. mentagrophytes* (5.5%), *M. canis* (4.5%), *E. floccosum* (1.3%), *N. gypsea* (0.4%), *M. audouinii* (0.05%), *T. verrucosum* (0.05%), and *Trichophyton* spp. (10.1%). These results also agree with the data obtained in the present study.

Finally, the three isolates from the Dominican Republic were identified as *T. rubrum/T. violaceum*, *T. mentagrophytes*/*T. interdigitale*, and *T. tonsurans*. The species *T. tonsurans* coincides with what was reported in other studies conducted in the Dominican Republic, where the species associated with *Tinea capitis* were *M. canis*, *M. audouinii*, and *T. tonsurans* [83,84].

The varying frequency of dermatophytes identified in this work can be explained by the evolution of urban and rural populations, the growing number of companion animals, and humidity and temperature, among other factors. These factors influence dermatophytes’ phenotypic and genotypic diversity in different geographical areas [51].

## 5. Conclusions

The present study confirms that identifying dermatophytes using conventional methods is not enough to obtain reliable results, because the morphology of these species is highly variable and they may present pleomorphism. In addition, adequate colony growth takes a long time, complicating their identification. Thus, a combination of nuclear gene sequence information can resolve several dermatophyte species’ boundaries and relationships, demonstrating this multigene approach’s advantages. The molecular markers used in the present work, ITS, *BT2*, and *Tef-1α*, were suitable, and the most informative locus was *Tef-1α*. On the other hand, optimal phylogenetic relationships were obtained with the concatenated sequences of the three markers through the Bayesian inference method.

## Figures and Tables

**Figure 1 jof-09-00462-f001:**
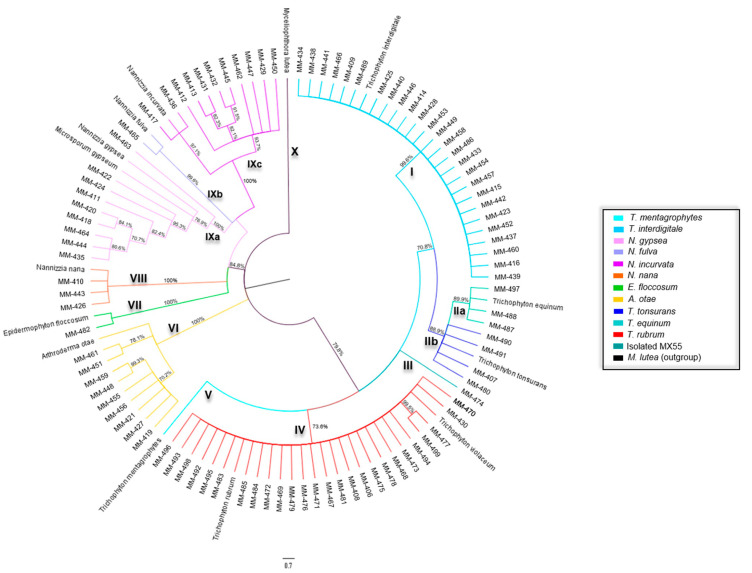
The phylogenetic tree was constructed using concatenated DNA sequences of markers ITS, *BT2*, and *Tef-1α* of dermatophyte species based on maximum likelihood method using MEGA X. Bootstrap support values are displayed at nodes.

**Figure 2 jof-09-00462-f002:**
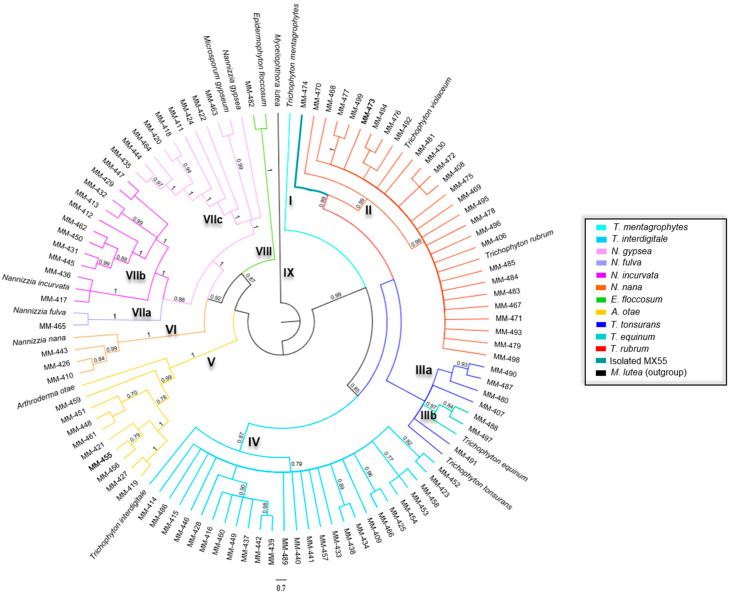
The phylogenetic tree was constructed using concatenated DNA sequences of ITS, *BT2*, and *Tef-1α* markers of dermatophyte species based on Bayesian inference using the MrBayes program with one million generations. Supporting values of posterior probability are displayed on nodes.

**Table 1 jof-09-00462-t001:** Epidemiological data of dermatophyte isolates included in the study.

Isolate	Geographical Origin	Gender	Edge	Clinical Form	Isolate	Geographical Origin	Gender	Edge	Clinical Form
MM-410	GT	M	32	*Tinea cruris*	MM-457	GT	M	48	*Tinea unguium*
MM-411	GT	F	31	*Tinea corporis*	MM-458	GT	F	25	*Tinea pedis*
MM-412	GT	M	2	*Tinea facei*	MM-459	GT	F	7	*Tinea capitis*
MM-413	GT	F	8	*Tinea corporis*	MM-460	GT	F	46	*Tinea pedis*
MM-414	GT	M	42	*Tinea corporis*	MM-461	GT	M	4	*Tinea capitis*
MM-415	GT	F	60	*Tinea unguim*	MM-462	GT	F	55	*Tinea corporis*
MM-416	GT	M	54	*Tinea unguim*	MM-463	GT	M	29	*Tinea corporis*
MM-417	GT	F	52	*Tinea capitis*	MM-464	GT	M	35	*Tinea manum*
MM-418	GT	F	48	*Tinea unguim*	MM-468	MX	M	35	*Tinea corporis*
MM-419	GT	M	3	*Tinea capitis*	MM-469	MX	F	17	*Tinea pedis*
MM-420	GT	F	41	*Tinea facei*	MM-470	MX	F	48	*Tinea unguium*
MM-421	GT	F	5	*Tinea capitis*	MM-471	MX	M	44	*Tinea unguium*
MM-422	GT	F	6	*Tinea capitis*	MM-472	MX	F	51	*Tinea unguium*
MM-423	GT	F	48	*Tinea unguium*	MM-473	MX	M	63	*Tinea pedis*
MM-424	GT	M	5	*Tinea capitis*	MM-474	MX	F	2	*Tinea capitis*
MM-425	GT	F	55	*Tinea corporis*	MM-475	MX	F	35	*Tinea unguium*
MM-426	GT	F	48	*Tinea pedis*	MM-476	MX	F	68	*Tinea pedis*
MM-427	GT	F	6	*Tinea capitis*	MM-477	MX	F	45	*Tinea corporis*
MM-428	GT	F	56	*Tinea pedis*	MM-478	MX	M	17	*Tinea unguium*
MM-429	GT	F	26	*Tinea corporis*	MM-479	MX	F	74	*Tinea unguium*
MM-430	GT	F	29	*Tinea unguium*	MM-480	MX	F	12	*Tinea capitis*
MM-431	GT	F	8	*Tinea capitiis*	MM-481	MX	F	37	*Tinea capitis*
MM-432	GT	F	22	*Tinea manum*	MM-482	MX	M	57	*Tinea pedis*
MM-433	GT	F	61	*Tinea unguium*	MM-483	MX	M	70	*Tinea corporis*
MM-434	GT	F	46	*Tinea pedis*	MM-484	MX	F	25	*Tinea capitis*
MM-435	GT	F	7	*Tinea facei*	MM-485	MX	M	48	*Tinea unguium*
MM-436	GT	M	5	*Tinea facei*	MM-486	MX	F	12	*Tinea capitis*
MM-437	GT	M	6	*Tinea facei*	MM-487	MX	M	6	*Tinea capitis*
MM-438	GT	M	48	*Tinea unguim*	MM-488	MX	F	4	*Tinea capitis*
MM-439	GT	M	57	*Tinea unguium*	MM-489	MX	F	2	*Tinea capitis*
MM-440	GT	F	64	*Tinea unguium*	MM-490	MX	M	8	*Tinea capitis*
MM-441	GT	F	57	*Tinea unguium*	MM-491	MX	F	11	*Tinea capitis*
MM-442	GT	F	36	*Tinea unguium*	MM-492	MX	F	64	*Tinea unguium*
MM-443	GT	F	42	*Tinea cruris*	MM-493	MX	F	34	*Tinea cruris*
MM-444	GT	F	52	*Tinea corporis*	MM-494	MX	F	74	*Tinea unguium*
MM-445	GT	F	56	*Tinea corporis*	MM-495	MX	F	59	*Tinea unguium*
MM-446	GT	F	68	*Tinea unguium*	MM-496	MX	F	8	*Tinea corporis*
MM-447	GT	F	4	*Tinea facei*	MM-497	MX	F	10	*Tinea capitis*
MM-448	GT	M	6	*Tinea capitis*	MM-498	MX	M	43	*Tinea unguium*
MM-449	GT	F	67	*Tinea unguium*	MM-499	MX	M	44	*Tinea unguium*
MM-450	GT	M	29	*Tinea corporis*	MM-407	DR	ND	ND	*Tinea unguium*
MM-451	GT	M	7	*Tinea capitis*	MM-408	DR	ND	ND	*Tinea unguium*
MM-452	GT	M	48	*Tinea unguium*	MM-409	DR	ND	ND	*Tinea unguium*
MM-453	GT	F	39	*Tinea unguium*	MM-465	HN	ND	ND	*Tinea unguium*
MM-454	GT	F	48	*Tinea unguium*	MM-466	HN	ND	ND	*Tinea unguium*
MM-455	GT	F	7	*Tinea capitis*	MM-467	HN	ND	ND	*Tinea unguium*
MM-456	GT	F	6	*Tinea facei*	MM-406	CR	ND	ND	*Tinea unguium*

MX, Mexico; GT, Guatemala; HN, Honduras; DR, Dominican Republic; CR, Costa Rica; M, male; F, female; ND, not defined.

**Table 2 jof-09-00462-t002:** Conidia size of identified dermatophytes.

Species	$\mathrm{Microconidia}\;\left(\overset{-}{x}\right)\mu m\pm \mathrm{SD}$		$\mathrm{Macroconidia}\;\left(\overset{-}{x}\right)\mu m\pm \mathrm{SD}$		
	Length	Width	Length	Width	Locules
*T. rubrum*	3.5 ± 0.4	1.7 ± 0.2	-	-	-
*T. mentagrophytes*	2.8 ± 0.7	2.1 ± 0.6	18.6 ± 5.7	3.3 ± 0.9	2 to 7
*T. tonsurans*	3.7 ± 0.1	2.0 ± 0.2	-	-	-
*N. gypsea*	4.4 ± 0.7	2.0 ± 0.3	40.1 ± 2.9	11.1 ± 2.1	3 to 6
*M. canis*	4.0 ± 0.2	1.6 ± 0.3	50.4 ± 9.4	15.5 ± 3.6	5 to 10
*N. nana*	-	-	11.9 ± 1.1	8.5 ± 0.3	1 to 2
*E. floccosum*	-	-	22.5 ± 2.7	7.6 ± 1.1	2 to 4

**Table 3 jof-09-00462-t003:** Genotypic identification of dermatophyte isolates included in the study using maximum likelihood.

Group	Subgroups	Isolates and Reference Sequences Obtained from GenBank	Species
I		MM-449, MM-458, MM-486, MM-433, MM-454, MM-457, MM-415, MM-442, MM-423, MM-452, MM-437, MM-460, MM-416, MM-439, MM-434, MM-438, MM-441, MM-466, MM-409, MM-489, MM-425, MM-440, MM-446, MM-414, MM-428, MM-453, JN134005.1, JF731043.1, KM678149.1	*T. mentagrophytes* *T. interdigitale*
II	IIa	MM-497, MM-488, MM-487, JN134108.1, JF731092.1, KM678112.1	*T. equinum*
	IIb	MM-480, MM-407, MM-491, MM-490, JN134084.1, JF731074.1, KM678205.1	*T. tonsurans*
III		MM-474, MM-470, MM-430, MM-477, MM-499, MM-494, MM-473, MM-468, MM-478, MM-475, MM-406, MM-408, MM-481, MM-467, MM-471, MM-476, MM-479, MM-469, MM-472, MM-484, MM-485, MM-483, MM-495, MM-492, MM-498, MM-493, MM-496, JN134104.1, JF731090.1, KM678140.1, JN134037.1, JF731058.1, KM678202.1	*T. violaceum * *T. rubrum*
IV		Z98000.1, KT155546.1, KM678083.1	*T. mentagrophytes*
V		MM-461, MM-451, MM-459, MM-448, MM-455, MM-456, MM-421, MM-427, MM-419, AB193632.1, JF731100.1, JN662936.1	*A. otae*
VI		MM-482, JN134157.1, JF731127.1, KM678060.1	*E. floccosum*
VII		MM-410, MM-443, MM-426, JN134095.1, KT55593.1, KM678111.1	*N. nana*
VIII	VIIIa	MM-463, MM-422, MM-424, MM-411, MM-420, MM-418, MM-464, MM-444, MM-435, GU291264.1, JF731093.1, KT261752.1, JN134132.1, KT155427.1, KM678057.1	*M. gypseum* *N. gypsea*
	VIIIb	MM-465, F078472.1, KT155473.1,KM678155.1	*N. fulva*
	VIIIc	MM-450, MM-429, MM-447, MM-462, MM-445, MM-432, MM-431, MM-413, MM-412, MM-436, MM-417, MH378242.1, KT155503.1, MH512804.1	*N. incurvata*
IX		KM655312.1, KX977026.1, HQ871722.1	*M. lutea* (outgroup)

**Table 4 jof-09-00462-t004:** Genotypic identification of dermatophyte isolates included in the study by Bayesian inference.

Groups	Subgroups	Isolates and Reference Sequences Obtained from GenBank	Species
I		Z98000.1, KT155546.1, KM678083.1	*T. mentagrophytes*
II		MM-474, MM-470, MM-468, MM-477, MM-499, MM-473, MM-494, MM-476, MM-492, MM-481, MM-430, MM-472, MM-408, MM-475, MM-469, MM-495, MM-478, MM-496, MM-406, MM-485, MM-484, MM-483, MM-467, MM-471, MM-493, MM-479, MM-498, JN134037.1, JF731058.1, KM678202.1, JN134194.1, JF731090.1, KM678140.1	*T. rubrum* *T. violaceum*
III	IIIa	MM-490, MM-487, MM-480, MM-407, MM-491, JN134084.1, JF731074.1, KM678205.1	*T. tonsurans*
	IIIb	MM-488, MM-497, JN134108.1, JF731092.1, KM678112.1	*T. equinum*
IV		MM-452, MM-423, MM-458, MM-453, MM-454, MM-425, MM-466, MM-409, MM-434, MM-438, MM-433, MM-457, MM-441, MM-440, MM-489, MM-414, MM-486, MM-415, MM-446, MM-428, MM-416, MM-460, MM-449, MM-437, MM-442, MM-439, JN134005.1, JF731043.1, KM678149.1	*T. mentagrophytes* *T. interdigitale*
V		MM-459, MM-451, MM-448, MM-461, MM-421, MM-455, MM-456, MM-427, MM-419, AB193632, JF731100.1, JN662936.1	*A. otae*
VI		MM-443, MM-426, MM-410, JN134095.1, KT55593.1, KM678111.1	*N. nana*
VII	VIIa	MM-465, F078472.1, KT155473.1, KM678155.1	*N. fulva*
	VIIb	MM-447, MM-429, MM-432, MM-413, MM-412, MM-462, MM-450, MM-431, MM-445, MM-436, MM-417, MH378242.1, KT155503.1, MH512804.1	*N. incurvata*
	VIIc	MM-463, MM-422, MM-424, MM-411, MM-418, MM-420, MM-464, MM-444, MM-435, GU291264.1, JF731093.1, KT261752.1, JN134132.1, KT155427.1, KM678057.1	*M. gypseum* *N. gypsea*
VIII		MM-482, JN134157.1, JF731127, KM678060.1	*E. floccosum*
IX		KM655312.1, KX977026.1, HQ871722.1	*M. lutea* (outgroup)

**Table 5 jof-09-00462-t005:** Frequency of species genotypically identified using maximum likelihood according to their geographic origin.

Species	(No. Isolated/Frequency)				
	MX	GT	HN	DR	CR
*T. mentagrophytes/T. interdigitale*	2/6.25%	22/40%	1/33.33%	1/33.33%	0
*T. equinum*	3/9.37%	0	0	0	0
*T. tonsurans*	3/9.37%	0	0	1/33.33%	0
*T. violaceum/T. rubrum*	23/71.87%	1/1.81%	1/33.33%	1/33.33%	1/100%
*M. canis*	0	9/16.36%	0	0	0
*E. floccosum*	1/3.12%	0	033.33%	0	0
*N. nana*	0	3/5.45%	0	0	0
*N. gypsea*	0	9/16.36%	0	0	0
*N. fulva*	0	0	1/33.33%	0	0
*N. incurvata*	0	11/20%	0	0	0
TOTAL	32	55	3	3	1

MX, Mexico; GT, Guatemala; HN, Honduras; DR, Dominican Republic; CR, Costa Rica.

**Table 6 jof-09-00462-t006:** Frequency of species genotypically identified using Bayesian inference according to their geographic origin.

Species	(No. Isolated/Frequency)				
	MX	GT	HN	DR	CR
*T. violaceum/T. rubrum*	24/72.72%	0	1/33.33%	1/33.33%	1/100%
*T. tonsurans*	4/12.12%	0	0	1/33.33%	0
*T. equinum*	2/6.06%	0	0	0	0
*T. mentagrophytes/T. interdigitale*	2/6.06%	22/40.74%	1/33.33%	1/33.33%	0
*M. canis*	0	9/16.16%	0	0	0
*N. nana*	0	3/5.5%	0	0	0
*N. fulva*	0	0	1/33.33%	0	0
*N. incurvata*	0	11/20.37%	0	0	0
*N. gypsea*	0	9/16.16%	0	0	0
*E. floccosum*	1/3.03%	0	0	0	0
TOTAL	33	54	3	3	1

MX, Mexico; GT, Guatemala; HN, Honduras; DR, Dominican Republic; CR, Costa Rica.

## Data Availability

All data generated or analysed during this study are included in this published article (and its Appendix A).

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
