# Peer review of "Phenotypic and Genotypic Identification of Dermatophytes from Mexico and Central American Countries"

_jof, 2023, doi:10.3390/jof9040462_

Round 1

Reviewer 1 Report

Major concern

-         What was the logic of sampling based on? Why are so few numbers of isolates from Costa Rica and the Dominican Republic used? How these samples can be the representative of related geographic origins? A collection of 94 isolates from 4 countries cannot outline the epidemiological trend of dermatophytosis in mentioned areas.

-         The authors used "Rush Munro technique" to describe the micro-morphological properties of the isolates. I think it was better to use "Dalmau plate" technique for this purpose because in Rush Munro technique a tease mount slide is prepared which lead to the destruct of the structural morphology.

-         In the last paragraph of M&M (lines 199-200) it was mentioned that the sequences obtained with the BT2, TEF-1α, and ITS markers were deposited in GenBank (Supplementary Table 1). I wish the authors mention the accession numbers in Result section to make ease the reviewers or readers sequence analysis.

-         In my opinion the title is inappropriate because as the authors mentioned in last paragraph of the Introduction section, the work aimed to identify the species of clinical isolates from some countries in South America. They used partial sequencing of ITS-rDNA along with BT2 and Tef -1α which are used to determine the species boundaries in dermatophytes rather than types. I suggest to change the title as "geographic distribution of dermatophytes in South America identified with multilocus sequence typing or similar titles.

-         Lines 34-35: For instance, T. rubrum/T. violaceum did not differ with the ITS and BT2 markers but with the Tef-1α gene.

As far know, Trichophyton rubrum strains have relatively identical ITS sequences, and characteristic sites were found in both ITS1 and ITS2. In contrast, despite the presence of species-specific sequences in ITS1, T. violaceum strains have intraspecies diversity, especially in the number of TA motifs at the end of ITS2 region. ITS sequencing from T. violaceum strains by Rezaei-Matehkolaei et al., [ref 2] provided complementary support for this fact. They also showed that the length of the consensus sequence for BT2 of T. rubrum and T. violaceum was the same (794 nt), but the two species differed in two signature nucleotides, that is, transitions in positions 173 and 376. Based on the fact that no intraspecies variation was found among strains of each species, the mentioned SNPs can act as molecular autapomorphies for species assignment of T. rubrum and T. violaceum isolates.

I didn't had access to T. rubrum and T. violaceum ITS sequences accession numbers however I think some errors happened in sequences. If possible make sequencing or sequence analysis again or share the raw sequences for T. rubrum and T. violaceum.

Introduction

-         In lines 53-56 the authors stated that "In their anamorphic state, these fungi mainly belong to the genera Trichophyton, Microsporum, and Epidermophyton. However, the new taxonomy proposed by de Hoog et al. [5] integrates three new genera: Lophophyton, Paraphyton, and Ctenomyces. The teleomorph state of these fungi integrates the genera Arthroderma and Nannizzia"….. Based on De Hoog et al., 2017 (ref 5) dermatophytes are currently in 7 distinct genera of Trichophyton, Microsporum, Epidermophyton, Lophophyton, Paraphyton, Arthroderma and Nannizzia. Ctenomyces is not a dermatophyte genus. Please correct the sentences.

Results

-         In line 208-209 some M. gypseum and M. nanum isolates were identified. Base on the latest taxonomy M. gypseum is synonymized with Nannizzia gypsea. Likewise, M. nanum is equal to N. nana. Please correct all M. gypseum and M. nanum through the paper based on De Hoog et al., 2017.

-         Tables 3 and 4 are unclear and complicated. I recommend the authors to provide a table with columns containing the ITS, BT2 and Tef-1α fragment sizes, sequence differences and percentage identity between all identified species. You can find similar tables in ref 2 and 30.

Discussion

-         Line 382, please change A. otae to M. canis.

-         I found no regular sampling in M&M. How the authors compared the distribution profile of studied isolates with studies by Martinez et al., Garcia and Meléndez, Welsh et al., Borman et al., and ….?

Author Response

Reviewer 1

Major concern

What was the logic of sampling based on? Why are so few numbers of isolates from Costa Rica and the Dominican Republic used? How these samples can be the representative of related geographic origins? A collection of 94 isolates from 4 countries cannot outline the epidemiological trend of dermatophytosis in mentioned areas.

Answer: We appreciate the reviewer's comment and agree that the small number of isolates from Honduras, the Dominican Republic, and Costa Rica is insufficient to infer an epidemiological trend in these countries. However, as mentioned in the manuscript, epidemiological data are scarce in these countries, and we thought it essential to include them, despite the small number, to provide recent information, highlighting the importance of genotypic identification of these, considering that in most of these countries, the identification of isolates is carried out fundamentally through phenotypic methods.

The authors used "Rush Munro technique" to describe the micro-morphological properties of the isolates. I think it was better to use "Dalmau plate" technique for this purpose because in Rush Munro technique a tease mount slide is prepared which lead to the destruct of the structural morphology.

Answer: We appreciate your suggestion to describe the micromorphology of the isolates through the "Dalmau plate" technique; we will consider it in future works. In this work, the "Rush Munro" technique efficiently demonstrated the microscopic characteristics of the isolates included in the study since it was only used to make a presumptive identification of the dermatophytes. Subsequently, the microculture technique was used to make a more accurate micromorphological description.

In the last paragraph of M&M (lines 199-200) it was mentioned that the sequences obtained with the BT2TEF-1α, and ITS markers were deposited in GenBank (Supplementary Table 1). I wish the authors mention the accession numbers in Result section to make ease the reviewers or readers sequence analysis.

Answer: The accession numbers in the Genbank were added in the Results section.

In my opinion the title is inappropriate because as the authors mentioned in last paragraph of the Introduction section, the work aimed to identify the species of clinical isolates from some countries in South America. They used partial sequencing of ITS-rDNA along with BT2 and Tef -1α which are used to determine the species boundaries in dermatophytes rather than types. I suggest to change the title as "geographic distribution of dermatophytes in South America identified with multilocus sequence typing or similar titles.

Answer: The title was changed, as you suggested.

Lines 34-35: For instance, T. rubrum/T. violaceum did not differ with the ITS and BT2 markers but with the Tef-1α gene.

As far know, Trichophyton rubrum strains have relatively identical ITS sequences, and characteristic sites were found in both ITS1 and ITS2. In contrast, despite the presence of species-specific sequences in ITS1, T. violaceum strains have intraspecies diversity, especially in the number of TA motifs at the end of ITS2 region. ITS sequencing from T. violaceum strains by Rezaei-Matehkolaei et al., [ref 2] provided complementary support for this fact. They also showed that the length of the consensus sequence for BT2 of T. rubrum and T. violaceum was the same (794 nt), but the two species differed in two signature nucleotides, that is, transitions in positions 173 and 376. Based on the fact that no intraspecies variation was found among strains of each species, the mentioned SNPs can act as molecular autapomorphies for species assignment of T. rubrum and T. violaceum isolates.

I didn't had access to T. rubrum and T. violaceum ITS sequences accession numbers however I think some errors happened in sequences. If possible make sequencing or sequence analysis again or share the raw sequences for T. rubrum and T. violaceum.

Answer: Although with the ITS and BT2 markers, the trees show the same topology, however, with the Tef-1α gene, they are separated into two subgroups in the same branch. In other studies, it has been observed that these two species are phenotypically different but phylogenetically similar by the multilocus method, and through the comparison of the complete genome of T. rubrum and T. violaceum, they are considered to have recently evolved as two species. (see the manuscript in the discussion section).

The sequences obtained with the BT2 markers and alpha elongation factor will be released on March 18 of the current year, which is why you did not have access. To complement the information from the present study's findings, we analyzed the sequences included in the T. rubrum/T. violaceum, in which it was impossible to accurately discern whether the isolates corresponded to T. rubrum or T. violaceum, since, as mentioned before, the molecular studies carried out so far are not conclusive to separate both species.

The DNA polymorphism analysis of the 26 sequences included in this group was performed with the DnaSP program ver. 6 (Rozas et al. Mol. Biol. Evol. 2017, 34: 3299-3302), with each of the markers used in this study. The results obtained with the ITS marker showed: 25 polymorphic sites and 13 singleton variable sites in positions 210, 237, 242, 248, 252, 255, 288, 418, 499, 502, and 510. At the same time, the BT2 marker showed 39 polymorphic sites and 12 singleton variable sites at positions 134, 135, 136, 137, 450, 462, 526, 530, 554, 575, 581, and 591. Likewise, the DNA polymorphism of the 26 sequences included in this group, obtained with the alpha elongation factor marker, showed 28 polymorphic sites and 11 singleton variable sites in the positions 197, 198, 201, 227, 241, 276, 277, 292, 479, 485, and 703. In addition, we built phylogenetic trees using the Maximum Likelihood method, with the 26 isolates corresponding to the T.rubrum/T. violaceum, with each of the markers used in the study, and the results also evidenced the lack of definition between the two species, as observed in the following figures.

Phylogenetic tree constructed from DNA sequences of marker ITS of dermatophytes, based on the Maximum Likelihood method using the MEGA X program. Bootstrap support values are displayed at nodes.

Phylogenetic tree constructed from DNA sequences of marker  BT2 of dermatophytes, based on the Maximum Likelihood method using the MEGA X program. Bootstrap support values are displayed at nodes.

Phylogenetic tree constructed from DNA sequences of marker Tef-1α of dermatophytes, based on the Maximum Likelihood method using the MEGA X program. Bootstrap support values are displayed at nodes.

Introduction

In lines 53-56 the authors stated that "In their anamorphic state, these fungi mainly belong to the genera Trichophyton, Microsporum, and Epidermophyton. However, the new taxonomy proposed by de Hoog et al. [5] integrates three new genera: Lophophyton, Paraphyton, and Ctenomyces. The teleomorph state of these fungi integrates the genera Arthroderma and Nannizzia"….. Based on De Hoog et al., 2017 (ref 5) dermatophytes are currently in 7 distinct genera of Trichophyton, Microsporum, Epidermophyton, Lophophyton, Paraphyton, Arthroderma and Nannizzia. Ctenomyces is not a dermatophyte genus. Please correct the sentences.

Answer: Corrected in manuscript.

Results

In line 208-209 some M. gypseum and M. nanum isolates were identified. Base on the latest taxonomy M. gypseum is synonymized with Nannizzia gypsea. Likewise, M. nanum is equal to N. nana. Please correct all M. gypseum and M. nanum through the paper based on De Hoog et al., 2017.

Answer: They were corrected in the manuscript.

Tables 3 and 4 are unclear and complicated. I recommend the authors to provide a table with columns containing the ITS, BT2 and Tef-1α fragment sizes, sequence differences and percentage identity between all identified species. You can find similar tables in ref 2 and 30.

Answer: The content of tables 3 and 4 were modified.

Discussion

Line 382, please change A. otae to M. canis.

Answer: Corrected in manuscript.

I found no regular sampling in M&M. How the authors compared the distribution profile of studied isolates with studies by Martinez et al., Garcia and Meléndez, Welsh et al., Borman et al., and ….?

Answer: Two tables were included (Tables 5 and 6) that show the most frequently identified species in each country, which allows comparing the results of this work with the cited works.

Reviewer 2 Report

Dear authors,

I read your submission carefully and found it suitable for publishing in JoF after considering the following comments and suggestions:

1- The title is not informative. It must contain the main message of your work. As a major consideration, there is no link between clinic and lab. data in your work. No data is presented about the source of the isolates and their geographic distribution as well as patient details. These information are mandatory to reach your work in a standard level for publishing in JoF.

2- The text needs major revision by a native expert company or person for the English language because it contains badly used words and sentences and scientific errors. Many sentences are difficult to understand. 

3- Add the patient data (sex, age, site of infection, etc.) as a separate Table for the heading 3.1. in results section. The table should contains strain numbers and geographic origin of the described dermatophytes as well.

4- Please go through the lines 123-125. Is that the correct composition of Sabouraud agar medium ? 

5- Summarize the lines 126-133.

6- What is the mean of "monosporic cultures" in the line 134 ?

7- What is APD in line 136 ?

8- Include the full name of YEPD in line 152.

9- Two references 41 and 42 are not specific for the dermatophytes. Add two references of Heidemann et al. (Br. J. Dermatol 2010: 162, 282-295) and Taghipour et al. (Mycoses doi: 10.1111/myc.12993) for ITS genotyping and the reference of Salehi et al. (Front. Microbiol. 12:643509. doi: 10.3389/fmicb.2021.643509) for simultaneous TEF-1α and BT2 genotyping. Likewise, discuss the data of these new references in comparison with your results in the discussion.

10- Consider Table 1 as a supplementary Table.

11- You should discuss about geographic distribution of identified dermatophytes as well as comparative results of morphological and molecular identification methods in accurate detection of the genera and species. This helps the readers to know about the importance of your work and its novelty.

Author Response

Reviewer 2

Dear authors,

I read your submission carefully and found it suitable for publishing in JoF after considering the following comments and suggestions:

1- The title is not informative. It must contain the main message of your work. As a major consideration, there is no link between clinic and lab. data in your work. No data is presented about the source of the isolates and their geographic distribution as well as patient details. These information are mandatory to reach your work in a standard level for publishing in JoF.

Answer: The title was changed.

2- The text needs major revision by a native expert company or person for the English language because it contains badly used words and sentences and scientific errors. Many sentences are difficult to understand. 

Answer: Submitted to English edition on MDPI.

3- Add the patient data (sex, age, site of infection, etc.) as a separate Table for the heading 3.1. in results section. The table should contains strain numbers and geographic origin of the described dermatophytes as well.

Answer: A table (Table 1) was included with the requested information.

4- Please go through the lines 123-125. Is that the correct composition of Sabouraud agar medium ? 

Answer: The composition of the culture medium was corrected.

5- Summarize the lines 126-133.

Answer: The indicated paragraph was summarized, as you suggested.

6- What is the mean of "monosporic cultures" in the line 134 ?

Answer: They are cultures developed from a single spore to guarantee purity and authenticity.

7- What is APD in line 136 ?

Answer: Corrected in the text.

8- Include the full name of YEPD in line 152.

Answer: Corrected in the text.

9- Two references 41 and 42 are not specific for the dermatophytes. Add two references of Heidemann et al. (Br. J. Dermatol 2010: 162, 282-295) and Taghipour et al. (Mycoses doi: 10.1111/myc.12993) for ITS genotyping and the reference of Salehi et al. (Front. Microbiol. 12:643509. doi: 10.3389/fmicb.2021.643509) for simultaneous TEF-1α and BT2 genotyping. Likewise, discuss the data of these new references in comparison with your results in the discussion.

Answer: The works of Taghipour et al. [43] and Salehi et al. [41] were included; however, the reference to Heidemann et al. was not included because the primers described do not correspond to those used in this work. Data are discussed concerning Salehi et al. (lines 400-402).

10- Consider Table 1 as a supplementary Table.

Answer: Table 1 was included in the supplementary material, as you suggested.

11- You should discuss about geographic distribution of identified dermatophytes as well as comparative results of morphological and molecular identification methods in accurate detection of the genera and species. This helps the readers to know about the importance of your work and its novelty.

Answer: Regarding the geographic distribution of the species, it would be interesting to be able to define a pattern by geographic region; however, the frequencies of species found in this work may not represent the reality of their regional distribution since the number of isolates analyzed from each country is not the same. Therefore, we consider it prudent not to speculate.

Regarding comparing the pheno and genotypic identification, the following paragraph was added after line 350.

Round 2

Reviewer 1 Report

Thanks the authors to improve the paper based on my comments and suggestions  and change the manuscript title based on my suggestion

I just recommend (if possible) to delete the table 2 because it has no new data. Instead,  just add the accession numbers to the results.

Author Response

Comments and Suggestions for Authors

Thanks the authors to improve the paper based on my comments and suggestions  and change the manuscript title based on my suggestion

I just recommend (if possible) to delete the table 2 because it has no new data. Instead, just add the accession numbers to the results.

Answer: Table 2 was removed and the accession numbers were added in the results (see manuscript).

Reviewer 2 Report

Necessary corrections and amendments were done and your work is now qualified for publishing in JoF. 

Author Response

Thanks for your recomendation.